# Anti-Osteoporotic Effects of *n-trans*-Hibiscusamide and Its Derivative Alleviate Ovariectomy-Induced Bone Loss in Mice by Regulating RANKL-Induced Signaling

**DOI:** 10.3390/molecules26226820

**Published:** 2021-11-11

**Authors:** Hyung Jin Lim, Eun-Jae Park, Yeong-Seon Won, Seon Gyeong Bak, Sun Hee Cheong, Seung Woong Lee, Soyoung Lee, Seung-Jae Lee, Mun-Chual Rho

**Affiliations:** 1Immunoregulatory Material Research Center, Korea Research Institute of Bioscience and Biotechnology (KRIBB), Jeongeup 56212, Korea; lhjin@kribb.re.kr (H.J.L.); pej911029@kribb.re.kr (E.-J.P.); 8wonys8@kribb.re.kr (Y.-S.W.); tsk9096@kribb.re.kr (S.G.B.); lswdoc@kribb.re.kr (S.W.L.); sylee@kribb.re.kr (S.L.); 2Department of Marine Bio Food Science, Chonnam National University, Yeosu 59626, Korea; sunny3843@jnu.ac.kr

**Keywords:** *n-trans*-hibiscusamide, 4-*O*-(*E*)-feruloyl-*N-*(*E*)-hibiscusamide, osteoclastogenesis, OVX induced osteoporosis

## Abstract

Osteoporosis is characterized by the deterioration of bone structures and decreased bone mass, leading to an increased risk of fracture. Estrogen deficiency in postmenopausal women and aging are major factors of osteoporosis and are some of the reasons for reduced quality of life. In this study, we investigated the effects of *n-trans*-hibiscusamide (NHA) and its derivative 4-*O*-(*E*)-feruloyl-*N*-(*E*)-hibiscusamide (HAD) on receptor activator of nuclear factor kappa-Β (NF-κB) ligand (RANKL)-induced osteoclast differentiation and an ovariectomized osteoporosis mouse model. NHA and HAD significantly inhibited the differentiation of osteoclasts from bone marrow-derived macrophages (BMMs) and the expression of osteoclast differentiation-related genes. At the molecular level, NHA and HAD significantly downregulated the phosphorylation of mitogen-activated protein kinase (MAPK) signaling molecules. However, Akt and NF-κB phosphorylation was inhibited only after NHA or HAD treatment. In the ovariectomy (OVX)-induced osteoporosis model, both NHA and HAD effectively improved trabecular bone structure. C-terminal telopeptide (CTX), a bone resorption marker, and RANKL, an osteoclast stimulation factor, were significantly reduced by NHA and HAD. The tartrate-resistant acid phosphatase (TRAP)-stained area, which indicates the osteoclast area, was also decreased by these compounds. These results show the potential of NHA and HAD as therapeutic agents for osteoporosis.

## 1. Introduction

Bone metabolism is maintained and regulated by osteoclasts and osteoblasts [1]. Osteoclasts and osteoblasts regulate bone resorption and bone formation. The activity and differentiation of osteoblasts and osteoclasts are regulated by hormones or cytokines such as 1,25-dihydroxyvitamin D3, thyroxine, parathyroid hormone (PTH), macrophage colony-stimulating factor (M-CSF) and receptor activator of nuclear factor kappa-Β (NF-κB) ligand (RANKL) [2,3]. M-CSF and RANKL are released from osteoblasts and affect the proliferation and differentiation of osteoclasts [4]. Osteoprotegerin (OPG) is also released from osteoblasts and inhibits the binding of RANK and RANKL [5]. Osteoclast differentiation is initiated by the binding of RANKL and RANK [5,6]. This binding recruits TNF receptor-associated factor 6 (TRAF6) and leads to the activation of Akt, mitogen-activated protein kinase (MAPK), NF-κB, and nuclear factor of activated T cells cytoplasmic 1 (NFATc1) signaling [7]. NFATc1 regulates osteoclast differentiation-related genes such as tartrate-resistant acid phosphatase (TRAP), cathepsin K (Ctsk), NFATc1, and matrix metalloproteinase-9 (MMP-9) [8,9].

Osteoporosis is characterized by the deterioration of bone structures and decreased bone mass, which lead to an increased risk of fracture [10]. There are two reasons for osteoporosis: estrogen deficiency in postmenopausal women and aging-related dietary calcium and vitamin D uptake [11,12]. Osteoporosis has become a major global health problem in an increasingly aging society [10]. Thus, various anti-osteoporosis agents have been developed, such as bisphosphonates and denosumab [13,14]. The effects of these agents are mainly focused on downregulating the bone resorption process. However, these treatments have severe side effects [15,16]. Therefore, phytochemicals, which have no side effects, may be good candidates for the development of new anti-osteoporosis agents.

In a previous study, we isolated several feruloyl amides, including *n-trans*-hibiscusamide (NHA), from *Portulaca oleracea*. Interleukin-6 (IL-6)/signal transducer and activator of transcription 3 (STAT3) inhibitory activity was the only reported biological activity of NHA [17]. IL-6, IL-6/STAT3 signaling mediates various diseases, such as inflammatory bowel disease, Castleman disease, osteoarthritis, and osteoporosis [18,19,20,21]. In osteoporosis, IL-6/STAT3 signaling is closely related to the differentiation of osteoclasts and RANKL expression in osteoblasts, which are regulators of bone homeostasis [22,23].

In this study, we investigated the effects of NHA and its derivative 4-*O*-(*E*)-feruloyl-*N*-(*E*)-hibiscusamide (HAD), which is obtained from the NHA synthesis process, on osteoclast differentiation in bone marrow-derived macrophages (BMMs) and showed the anti-osteoporotic effects of these compounds using an ovariectomy (OVX)-induced osteoporosis mouse model.

## 2. Results

### 2.1. NHA and HAD Inhibit RANKL-Induced Osteoclast Differentiation

Osteoclasts are differentiated from BMMs and exhibit bone resorption activity. TRAP is a marker of mature osteoclasts and is often used to identify osteoclasts. To examine whether NHA and HAD (Figure 1A) inhibit RANKL-induced osteoclastogenesis, an osteoclast differentiation assay using TRAP staining was performed. TRAP-positive multinucleated cells with more than three nuclei were counted. The number of TRAP-positive multinucleated cells significantly decreased in a dose-dependent manner in response to NHA and HAD (Figure 1B,C). To confirm whether this result was due to cytotoxicity, an XTT assay was performed. However, there was no cytotoxicity (Figure 1D).

### 2.2. NHA and HAD Inhibit RANKL-Induced Osteoclast Differentiation-Related Gene Expression

NFATc1 forms a complex with activator protein 1 (AP-1), which is an essential transcription factor associated with osteoclastogenesis, to efficiently induce osteoclast-specific genes such as Ctsk, dendrocyte-expressed seven transmembrane protein (DC-STAMP), MMP-9, and NFATc1 [24,25,26]. To confirm that NHA and HAD affect the transcription factor NFATc1 and the expression of osteoclast-specific genes, quantitative real-time RT-PCR analysis was performed. NHA and HAD significantly decreased all osteoclast differentiation genes in a dose-dependent manner (Figure 2).

### 2.3. NHA and HAD Inhibit RANKL-Induced Osteoclast Differentiation-Associated Signaling Molecules

Osteoclast differentiation is initiated by RANKL and RANK binding. Then, TRAF6 is recruited and activates downstream signaling pathways, including MAPK, Akt, and NF-κB. The activation of these signaling pathways upregulates NFATc1 expression, which is related to osteoclast differentiation. To confirm that NHA and HAD affect these signaling molecules, Western blot analysis was performed. The phosphorylation of MAPKs, including ERK, JNK, and p38, was significantly downregulated by NHA and HAD at one time point at least (Figure 3). However, the phosphorylation of NF-κB p65 and Akt was significantly downregulated only after NHA or HAD treatment (Figure 3).

### 2.4. NHA and HAD Alleviate OVX-Induced Bone Loss

To investigate the effects of NHA and HAD on an in vivo model, we used an OVX-induced bone loss model. The OVX model mimics estrogen deficiency in postmenopausal women and induces osteoporosis. The experimental procedure is shown in Figure 4A, and alendronate was used as positive control. After the experiment, mouse femurs were analyzed using micro-computed tomography (CT) to confirm the effects of NHA and HAD on the cancellous bone matrix of the OVX-induced model. The micro-CT image results showed that the trabecular bone structures were restored by 30 mg/kg NHA and HAD compared with those in the OVX-only group (Figure 4B). The bone volume/total volume (BV/TV) ratio, trabecular thickness (Tb.Th), trabecular number (Tb.N), and trabecular bone mineral density (BMD) parameters, which indicate trabecular bone status, were significantly restored by 30 mg/kg NHA and HAD (Figure 4C).

### 2.5. NHA and HAD Downregulate Biochemical Markers of Osteoporosis in an OVX-Induced Bone Loss Model

There are various biochemical markers in serum. For example, RANKL is released from osteoblasts and differentiates preosteoclasts into osteoclasts. OPG is also secreted by osteoblasts and inhibits RANKL binding to RANK. C-terminal telopeptide (CTX) is a fragment of collagen that composes the bone matrix and is often used as a bone resorption marker [27]. To investigate the effects of NHA and HAD on the OVX model, serum biochemical markers were analyzed by ELISA. The results showed that NHA and HAD decreased CTX compared with that in the OVX group. This finding suggests that the compounds effectively inhibited bone resorption by osteoclasts (Figure 5A). RANKL was significantly downregulated by NHA and HAD treatment; however, OPG was not affected (Figure 5B,C). Finally, the RANKL/OPG ratio was significantly decreased by NHA and HAD treatment (Figure 5D).

## 3. Discussion

Bone metabolism is regulated and maintained by osteoclasts and osteoblasts [1]. Imbalances in bone resorption by osteoclasts and bone formation by osteoblasts cause severe bone disease [1,28]. Estrogen deficiency in postmenopausal women breaks the balance between bone resorption and bone formation and causes osteoporosis [11,29]. Osteoporosis is characterized by low BMD and bone strength and increased bone fragility and risk of bone fracture [10]. Increasing bone formation by osteoblasts and decreasing bone resorption by osteoclasts are two major therapeutic strategies for treating osteoporosis [30,31].

Bone marrow monocytes, which are preosteoclasts, differentiated into BMMs and osteoclast precursor cells when stimulated with M-CSF. RANKL and M-CSF promote the differentiation of BMMs to mononuclear osteoclasts, which express TRAP. Then, osteoclast fusion occurs in response to various factors, such as c-Fos, NFATc1, DC-STAMP, and NF-κB, and multinucleated osteoclasts are formed. Finally, Ctsk, the proto-oncogene tyrosine protein kinase Src (c-Src) and integrin b3, lead the maturation of osteoclasts to form active osteoclasts [6,7,32]. In this study, NHA and HAD effectively inhibited RANKL-induced TRAP-positive multinucleated osteoclasts in vitro. HAD decreased the numbers of TRAP-positive and multinucleated cells; however, NHA inhibited only multinucleated cells. The expression of osteoclast differentiation-related genes, such as NFATc1, DC-STAMP, Ctsk, and MMP-9, was downregulated by NHA and HAD. HAD treatment inhibited the expression of more genes than NHA, except NFATc1. RANKL signaling has several downstream signaling pathways, including Akt, MAPK, NF-κB, and calcium signaling [6,7]. Akt signaling induces osteoclastogenesis by upregulating NFATc1 [33]. Among the MAPKs, the p38 signaling pathway plays a pivotal role in osteoclast formation. p38α knockout induces bone mass in young mice [34]. HAD inhibited RANKL-induced Akt, MAPK, and NF-κB signaling; however, NHA did not affect NF-κB. Furthermore, NHA strongly inhibited ERK and p38 phosphorylation compared with the effect of HAD. These results could explain the downregulation of NFATc1 by NHA, which was affected by Akt. In this study, the effect of NHA and HAD on Akt, MAPK, and NF-κB was investigated; however, the effect on calcium signaling was not. RANKL induces calcium signaling via PLCγ2 and plays a crucial role in the migration and adhesion of preosteoclasts [35]. Furthermore, the deletion of PLCγ2 downregulated RANKL-induced NFATc1 expression [36]. To identify the exact molecular mechanisms of NHA and HAD, their effects on calcium signaling should be studied.

In the OVX model, NHA and HAD improved bone microarchitecture. It could be explained by the fact that the compounds decrease osteoclast formation in bone matrix; then, bone resorption of trabecular bone was reduced, and the bone microarchitecture is recovered. The serum level of CTX data shows that the compounds treatment downregulates bone resorption in the OVX model. Furthermore, the serum level of RANKL, OPG, and the RANKL/OPG ratio also indicate that NHA and HAD affect the RANKL/OPG axis of the osteoblast and inhibit osteoclast formation. It is thought that the compounds inhibit JAK2 in osteoblasts and affect the expression of RANKL. Recent studies have revealed that IL-6/STAT3 signaling mediates osteoclast differentiation directly and indirectly [22,23]. Blocking IL-6/STAT3 downregulates RANKL expression in osteoblasts and inhibits osteoclast differentiation in preosteoclasts [37]. Yang et al. reported that STAT3 regulates NFATc1 expression in STAT3-deficient mice [22]. Furthermore, Wang et al. revealed that RANKL expression was induced by the IL-6/STAT3 signaling pathway [38]. Thus, the effects of NHA and HAD on RANKL-induced osteoclastogenesis and on OVX-induced osteoporosis in mice could be explained by the inhibition of IL-6/STAT3. Furthermore, we treated the compounds after 6 weeks after OVX, which means that the therapeutic effect of the compounds was investigated. In bone metabolism, the inhibition of bone resorption of osteoclasts could increase bone mass and strength. Thus, NHA and HAD could display not only therapeutic effect but also a protective effect on osteoporosis through inhibiting bone loss of initial stage of osteoporosis. However, there are some limitations in this study. To explain the anti-osteoporotic effect of NHA and HAD due to IL-6/STAT3 inhibition, the effects of NHA and HAD on STAT3 phosphorylation in response to RANKL stimulation and OVX models should be investigated. Thus, the effects of these compounds on osteoblasts should be further studied.

In summary, we showed the effects of NHA and HAD on RANKL-induced osteoclast differentiation. The compounds decreased mature osteoclasts and the expression of osteoclast differentiation-related genes. Western blot analysis showed that the compounds inhibited RANKL signaling molecules. In the OVX model, NHA and HAD alleviated bone loss and bone resorption-related markers. In conclusion, NHA and HAD inhibit RANKL-induced osteoclast differentiation and OVX-induced bone loss through the regulation of osteoclasts. Therefore, NHA and HAD could be potential therapeutic agents for treating osteoporosis.

## 4. Materials and Methods

### 4.1. Reagents

All cell culture reagents, including media, antibiotics, and fetal bovine serum (FBS), were purchased from Gibco BRL (Grand Island, NY, USA). Recombinant human M-CSF and human RANKL were purchased from Peprotech (London, UK). All antibodies were obtained from Cell Signaling Technology (Boston, MA, USA).

### 4.2. Synthesis of NHA and HAD

A mixture of 3,5-dimethoxy-4-hydroxyphenethylamine HCl (0.80 g, 3.4 mmol), ferulic acid (0.80 g, 4.1 mmol), and dimethylformamide (DMF, 24 mL) was dissolved at room temperature. To this solution, 1-ethyl-3-(3-dimethylaminopropyl)carbodiimide (EDC, 2.4 mL, 13.6 mmol) and triethanolamine (TEA, 1.8 mL) were added at −5 °C for 10 min, and then, the mixture was stirred at room temperature overnight. The solvent was removed in vacuo, and the residue was purified by a flash column chromatography instrument on a silica column (SiO_2_, 120 g, CHCl_3_:MeOH, 1:0 → 30:1, *v*/*v*) to generate 11 subfractions. Fraction 8 (868.2 mg) was rechromatographed on a reversed-phase C_18_ column using flash column chromatography (C_18_, 130 g, H_2_O:MeOH, 13:7 → 2:3, *v*/*v*) to yield NHA(168.5 mg) and its derivative HAD (52.7 mg), as determined by ^1^H, ^13^C, and 2D NMR (COSY, HMQC, HMBC, NOESY) and MS spectroscopic data.

*N*-(*E*)-*Hibiscusamide* (**1**), Yield 11%; White solid; ESI-MS: *m*/*z* 374 [M + H]^+^; ^1^H NMR (600 MHz, methanol-*d*_4_) *δ*_H_: 7.44 (1H, d, *J* = 15.6 Hz, H-7), 7.10 (1H, d, *J* = 1.8, H-2), 7.02 (1H, dd, *J* = 8.4, 1.8 Hz, H-6), 6.79 (1H, d, *J* = 8.4, H-5), 6.52 (2H, s, H-2′, 6′), 6.41 (1H, d, *J* = 15.6 Hz, H-8), 3.50 (1H, t, *J* = 7.2 Hz, H-8′), 2.77 (1H, t, *J* = 7.2 Hz, H-7′), 3.88 (3H, s, 3-OCH_3_), 3.82 (6H, s, 3′, 5′-OCH_3_); ^13^C NMR (150 MHz, methanol-*d*_4_) *δ*_C_: 169.3 (C-9), 150.0 (C-4′), 149.4 (C-3), 149.3 (C-3′, 5′), 142.2 (C-7), 135.2 (C-4′), 131.4 (C-1), 128.4 (C-1′), 123.3 (C-6), 118.9 (C-8), 116.6 (C-5), 111.7 (C-2), 107.2 (C-2′, 6′), 42.6 (C-8′), 36.8 (C-7′), 56.9 (3′, 5′-OCH_3_), 48.5 (3-OCH_3_). 

*HAD* (**2**), Yield 3%; White solid; ESI-MS: *m*/*z* 550 [M + H]^+^; ^1^H NMR (600 MHz, methanol-*d*_4_) δ_H_: 7.78 (1H, d, *J* = 16.2 Hz, H-7″), 7.52 (1H, d, *J* = 15.6 Hz, H-7), 7.26 (2H, m, H-2, 2″, overlap), 7.18 (1H, dd, *J* = 7.8, 1.8 Hz, H-6), 7.15 (1H, dd, *J* = 8.4, 1.8 Hz, H-6″), 7.10 (1H, d, *J* = 8.4 Hz, H-5″), 6.84 (1H, d, *J* = 8.4 Hz, H-5″), 6.58 (1H, d, *J* = 15.6 Hz, H-8), 6.58 (1H, d, *J* = 16.2 Hz, H-8″), 6.53 (2H, s, H-2′, 6′) 3.52 (1H, t, *J* = 7.8 Hz, H-8′), 2.79 (1H, t, *J* = 7.8 Hz, H-7′), 3.91 (3H, s, 3″-OCH_3_), 3.86 (3H, s, 3-OCH_3_), 3.83 (6H, s, 3′, 5′-OCH_3_); ^13^C NMR (150 MHz, methanol-*d*_4_) δ_C_: 168.6 (C-9), 167.2 (C-9″), 153.3 (C-3), 151.2 (C-4″), 149.6 (C-3″), 149.4 (C-3′, 5′), 148.9 (C-7″), 142.7 (C-4), 141.2 (C-7), 135.4 (C-1), 135.2, (C-4′), 131.3 (C-1′), 127.7 (C-1″), 124.7 (C-6″), 124.5 (C-5), 122.3 (C-8), 121.7 (C-6), 116.7 (C-5″), 114.3 (C-8″), 112.7 (C-2), 112.1 (C-2″), 107.2 (C-2′, 6″), 42.6 (C-8′), 36.8 (C-7′), 56.9 (3′, 5′-OCH_3_), 56.6 (3″-OCH_3_), 56.5 (3-OCH_3_); HRESI-TOF-MS: *m*/*z* 548.1909 [M − H]^−^ (calcd. for C_30_H_32_NO_9_, 548.1926).

### 4.3. Isolation of Bone Marrow Monocytes and the Differentiation of BMMs

The mouse femurs and tibias were obtained from 5-week-old ICR mice and were placed in ice-cold α-MEM supplemented with 10% FBS, 100 U/mL penicillin, and 100 mg/mL streptomycin. The tips of the bones were removed, and bone marrow was flushed out with α-MEM using a syringe. The obtained bone marrow was passed through a 4 μm nylon mesh filter, and red blood cell lysis buffer was added to destroy red blood cells. The remaining cells were treated with M-CSF (30 ng/mL) for 3 days to differentiate them into BMMs.

### 4.4. In Vitro Osteoclastogenesis Assay

BMMs were plated on 48-well culture plates at a density of 2 × 10^4^ cells/well and treated with M-CSF (30 ng/mL) and RANKL (100 ng/mL) after being pretreated with various concentrations of NHA and HAD. DMSO and phosphate-buffered saline (PBS) were used as vehicles for the compounds, M-CSF and RANKL. After 5 days, the cells were fixed and stained for TRAP according to the manufacturer’s instructions (Sigma-Aldrich, St. Louis, MO, USA). In brief, the cells were fixed with 10% formaldehyde and washed with deionized water. Then, the samples were stained with the TRAP solution for 1 h. The TRAP solution was prepared according to the manufacturer’s instructions. After being stained, the cells were washed, and TRAP-positive multinucleated osteoclasts were counted.

### 4.5. Cell Viability Assay

BMMs were seeded in 96-well plates at a density of 1 × 10^4^ cells/well. The cells were treated with various concentrations of NHA and HAD in the presence of M-CSF (30 ng/mL) for 72 h. DMSO and PBS were used as vehicles for the compounds and M-CSF. Then, 50 µL of XTT solution (Cell Signaling Technology) was added to the culture plate and incubated for 3 h. The absorbance was measured at 450 nm using a microplate ELISA reader (Molecular Devices, Sunnyvale, CA, USA).

### 4.6. Western Blot Analysis

BMMs were pretreated with NHA and HAD for 1 h and then stimulated with RANKL (100 ng/mL) for the indicated times. The cells were harvested, and total proteins were extracted. Then, the proteins were loaded onto 4–12% SDS-PAGE gels and separated. Next, the proteins were transferred onto polyvinylidene fluoride (PVDF) membranes and blocked with tris-buffered saline (TBS) containing 5% BSA. After being blocked, the membranes were washed and incubated with the appropriate primary and secondary antibodies. Finally, the membranes were developed using a West-Queen RTS Western Blot Detection Kit (iNtRON Bio. Seongnam, Korea).

### 4.7. Quantitative Real-Time RT-PCR

BMMs were seeded in 6-well plates and treated with RANKL (100 ng/mL) for 48 h after being pretreated with 10, 30, and 60 μM NHA and HAD for 1 h. The cells were harvested, and total RNA was isolated by using a PureLink RNA Mini Kit (Ambion, Foster City, CA, USA) according to the manufacturer’s protocol. Then, complementary DNA (cDNA) was synthesized using SuperScript III First Strand Synthesis SuperMix (Invitrogen). Real-time PCR was performed with a StepOnePlus Real-Time PCR System (Applied Biosystems, Foster City, CA, USA) and TaqMan Gene Expression Master Mix, appropriate TaqMan Gene assay primers and Eukaryotic 18S rRNA Endogenous Control (Applied Biosystems). The gene expression levels were normalized to the level of human 18S rRNA.

### 4.8. Induction and Treatment of Osteoporosis

Detailed procedures are described in a previous study [39]. In brief, six-week-old female C57BL/6 mice were purchased from OrientBio (Seongnam, Korea). After being acclimatized for one week, the mice underwent a sham operation or bilateral OVX under anesthesia. The mice were maintained without any treatment for 6 weeks after surgery until bone loss was apparent. After 6 weeks, NHA, HAD, and alendronate were dissolved in 10% DMSO containing PBS and intraperitoneally injected into the appropriate groups every 2 days for 6 weeks. The mice were randomly divided into 6 groups (n = 6 mice per group): sham-operated mice (SHAM), OVX mice treated with vehicle (OVX), OVX mice treated with 2 mg/kg alendronate (ALN), OVX mice treated with NHA (10 mg/kg or 30 mg/kg), and OVX mice treated with HAD (10 mg/kg or 30 mg/kg). Alendronate was used as positive control. The animal experimental protocols were approved by the Institutional Animal Care and Use Committee of Korea Research Institute of Bioscience and Biotechnology (permission number # KRIBB-AEC-20149), and all mice were handled in accordance with the Guide for the Care and Use of Laboratory Animals published by the US National Institutes of Health.

### 4.9. Micro-CT Measurements

Detailed procedures are described in a previous study [39]. In brief, the femurs were removed from the mice, the soft tissue was removed, and the bones were fixed in 10% formalin and stored in PBS at 4 °C. Bone morphometric parameters and microarchitectural properties of the femurs were analyzed using a SkyScan 1076 micro-CT scanner (Bruker microCT, Kontich, Belgium). The femurs were placed in polystyrene tubes filled with PBS to prevent drying during scanning. Each specimen was mounted on the 1076 scanner sample chamber for micro-CT imaging, and the sample was rotated automatically along the axis (angular step: 0.42°, reconstruction angular range: 360.36°). Scanning was performed using a 0.5 mm Al filter, an 88 kV source voltage, and a 112 μA source current with 9 μm resolution. The region of interest was defined between 0.5 and 2.5 mm below the growth plate of the proximal femur. The BV/TV ratio, Tb.N, Tb.Th and three-dimensional images of the femur were analyzed using the Nrecon^®^, CTAn^®^ and CTVol^®^ software.

### 4.10. Biochemical Analysis of Serum

Blood samples were collected from the mice by cardiac puncture before the mice were sacrificed. Serum was obtained by centrifugation and then stored at −80 °C for the analysis of biochemical parameters. The serum levels of CTX, OPG, and RANKL were measured by commercial ELISA kits according to the manufacturer’s instructions. The ELISA kits for OPG and RANKL were obtained from Abcam, and the CTX ELISA kit was obtained from Immunodiagnostic Systems (Boldon, UK).

### 4.11. Statistical Analysis

The results are shown as the mean ± standard deviation (SD) of three or six independent experiments. Statistical analysis was performed using Prism 5 software (GraphPad Software, San Diego, CA, USA). Statistical significance was calculated using one-way ANOVA followed by Dunnett’s test or unpaired Student’s *t*-test.

## Figures and Tables

**Figure 1 molecules-26-06820-f001:**
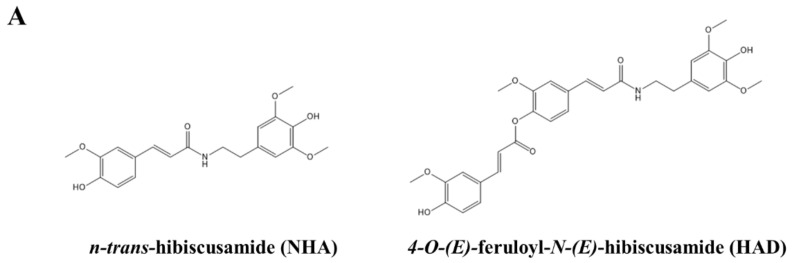
Inhibitory effect of NHA and HAD on osteoclast differentiation and the cytotoxicity of NHA and HAD. (**A**) Structures of NHA and HAD. (**B**) BMMs were cultured with M-CSF (30 ng/mL) and RANKL (100 ng/mL) in the presence of the indicated concentrations of NHA and HAD for 5 days. Then, the cells were fixed and stained with the TRAP staining solution, and (**C**) TRAP-positive multinucleated cells (≥3 nuclei) were counted. (**D**) BMMs were seeded in 96-well plates and treated with the indicated concentrations of NHA and HAD in the presence of M-CSF (30 ng/mL) for 3 days. Cell viability was determined by XTT assays. The results are presented as the mean ± standard deviation (SD) of three individual experiments. * *p* < 0.05, ** *p* < 0.01 compared with the control group.

**Figure 2 molecules-26-06820-f002:**
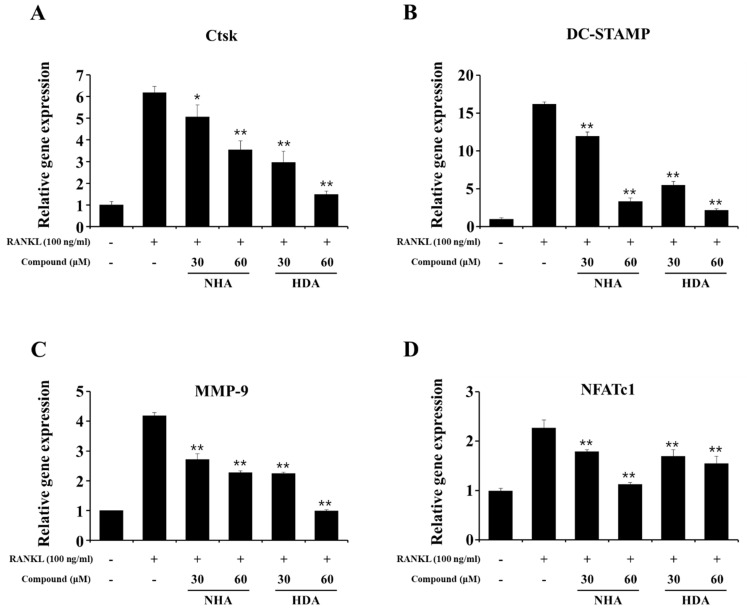
Inhibitory effect of NHA and HAD on RANKL-induced osteoclast-specific genes. BMMs were seeded in six-well plates and treated with RANKL (100 ng/mL) for 48 h after being pretreated with the indicated concentrations of NHA and HAD for 1 h. Then, the cells were harvested, and the gene expression levels of (**A**) Ctsk, (**B**) DC-STAMP, (**C**) MMP-9, and (**D**) NFATc1 were analyzed by quantitative real-time RT-PCR. The results are presented as the mean ± SD of three individual experiments. * *p* < 0.05, ** *p* < 0.01 compared with the RANKL-only group.

**Figure 3 molecules-26-06820-f003:**
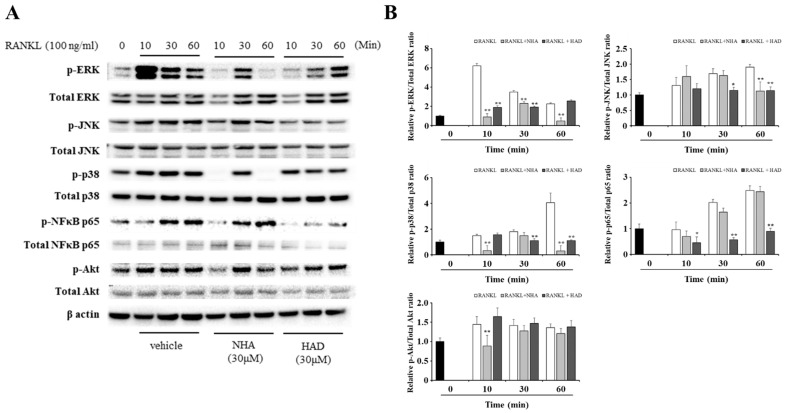
Inhibitory effect of NHA and HAD on the RANKL-induced signaling pathway. (**A**) BMMs were seeded in six-well plates and treated with RANKL (100 ng/mL) for the indicated times after being pretreated with 30 μM NHA and HAD. Then, the cells were collected and subjected to Western blot analysis using the indicated antibodies. (**B**) The optical densities of the bands were calculated by ImageJ software. The results are presented as the mean ± SD of three individual experiments. * *p* < 0.05, ** *p* < 0.01 compared with the RANKL-only group.

**Figure 4 molecules-26-06820-f004:**
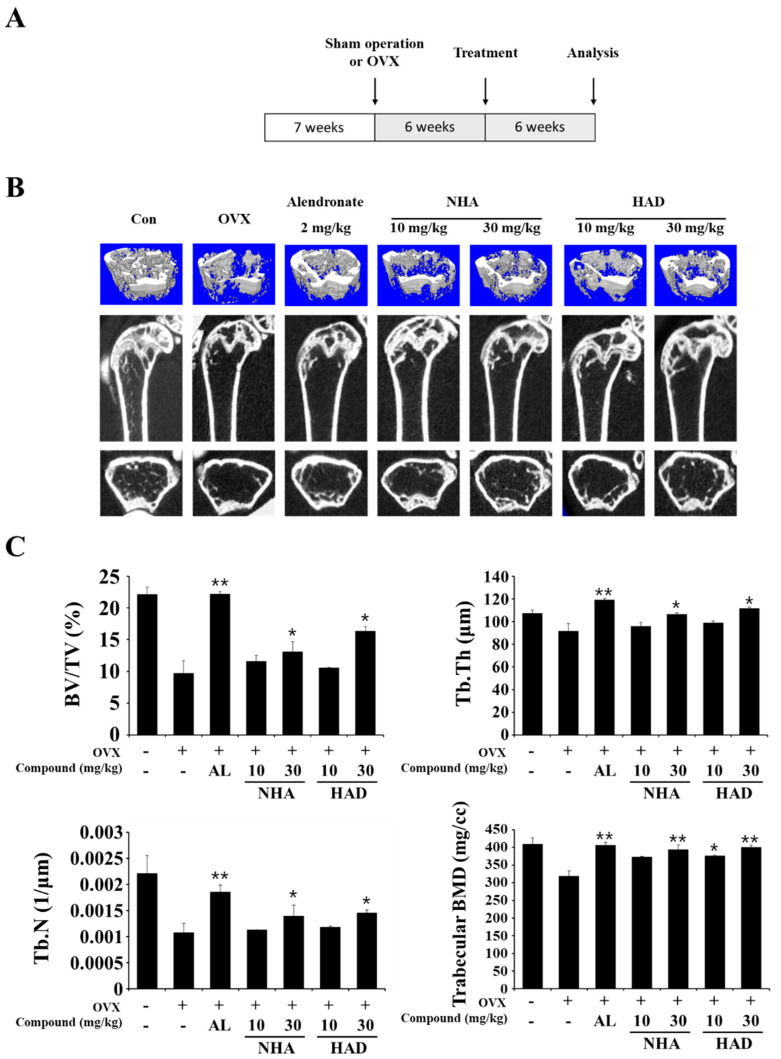
(**A**) Schematic diagram illustrating the experimental design of the OVX-induced osteoporosis model. (**B**,**C**) Micro-CT analysis of the proximal femurs of NHA-, HAD-, or alendronate-treated ovariectomized mice. (**B**) Micro-CT images of longitudinal and transverse proximal femurs from sham-operated, OVX + vehicle-, OVX + NHA-treated, OVX + HAD-treated, and OVX + alendronate-treated mice were obtained. (**C**) The bone volume/total volume (BV/TV), trabecular thickness (Tb.Th), trabecular number (Tb.N), and trabecular bone mineral density (BMD) of the femurs were analyzed using micro-CT. The data are expressed as the means ± SD * *p* < 0.05, ** *p* < 0.01 compared with the OVX-only group.

**Figure 5 molecules-26-06820-f005:**
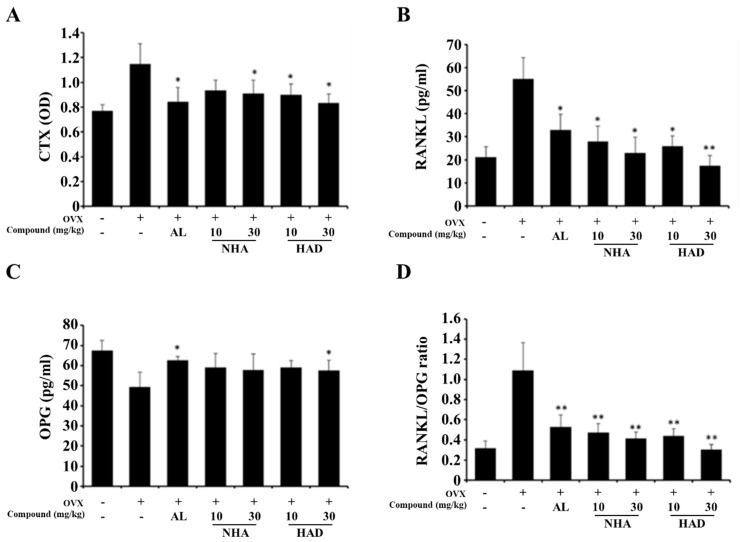
Biochemical analysis of NHA- and HAD-treated ovariectomized mice. The serum levels of (**A**) CTX, (**B**) RANKL, (**C**) OPG, and (**D**) RANKL and the OPG ratio were analyzed by ELISA. The data are expressed as the means ± SD * *p* < 0.05, ** *p* < 0.01 compared with the OVX-only group.

## Data Availability

Not applicable.

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
