# Peer review of "Anti-Osteoporotic Effects of *n-trans*-Hibiscusamide and Its Derivative Alleviate Ovariectomy-Induced Bone Loss in Mice by Regulating RANKL-Induced Signaling"

_molecules, 2021, doi:10.3390/molecules26226820_

Round 1

Reviewer 1 Report

In this study Hyung-Jin Lim and collaborators aim to evaluate the effects of NHA and its derivative HAD on receptor activator of nuclear factor kappa-Β (NF-κB) ligand 19 (RANKL)-induced osteoclast differentiations. Moreover, an ovariectomized osteoporosis mouse model was also employed for this study. The study is well designed, and data well shown by using an in vitro model and a mouse model of osteoporosis.

The content of the manuscript is very interesting with clear results. Introduction and discussion are clear and focused on the topic. Methods and results are well concise in text and figures.

Only the following issues should be addressed in order to improve the manuscript:

  1. Page 2 line 73, “Osteoclasts differentiate from BMMs and” something is missing.
  2. Figure 5. The resolution of the graph is not good, graph C is better than the others. Please improve the resolution. Same for graphs of figure 4.
  3. Line 169 of discussion, please better write this sentence.

Author Response

Reviewer 1

In this study Hyung-Jin Lim and collaborators aim to evaluate the effects of NHA and its derivative HAD on receptor activator of nuclear factor kappa-Β (NF-κB) ligand 19 (RANKL)-induced osteoclast differentiations. Moreover, an ovariectomized osteoporosis mouse model was also employed for this study. The study is well designed, and data well shown by using an in vitro model and a mouse model of osteoporosis.

The content of the manuscript is very interesting with clear results. Introduction and discussion are clear and focused on the topic. Methods and results are well concise in text and figures.

Only the following issues should be addressed in order to improve the manuscript:

Point 1) line 73, “Osteoclasts differentiate from BMMs and” something is missing.

Response 1) Thanks for your comment. We revised it.

Point 2) Figure 5. The resolution of the graph is not good, graph C is better than the others. Please improve the resolution. Same for graphs of figure 4.

Response 2) Thanks for your review. We fixed it.

Point 3) Line 169 of discussion, please better write this sentence.

Response 3) Thanks for your review. We revised the sentence.

Reviewer 2 Report

Lim et al. explored the role of NHA and HAD in inhibiting osteoclastogenesis in this manuscript. The topic is quite interesting and the structure of the study is clear and easy to understand. However, I have several suggestions to strengthen the rationality of this manuscript.

  1. In the previous article, the author analyzed 10 different compounds why choose the compound 6? For example, the most significant drug in inhibiting ERK pathway activation was compound 5. In addition, what is the structure of HAD, a derivative of NHA, and is there any previous literature discussing its activity?

  1. Fig.1a, this picture is fuzzy, and it is suggested that the author should place a higher magnification and higher resolution figure for presentation.

  1. The definition of TRAP-positive multinucleated cell (line 76-77) should be placed in the figure legend.

  1. Besides the differentiation of osteoclast as shown in Figure 1, do NHA and HAD have any effect on actin ring formation?

  1. In Figure 3A, why is the p-Akt basal level of BMMs so high without RANKL?

  1. In FIG. 3B, at 0 h, why are the expressions of all proteins high in the RANKL+HAD group? Is there a group without RANKL as a control group for protein expression?

  1. Alendronate appears in figure 4, but is not explained in the manuscript.

  1. It is not appropriate to introduce experimental results in detail in the discussion section (Line 189-198...). The authors may describe whether there are other related compounds in clinical use for osteoporosis prevention or the expected effectiveness of the compound discussed here compared to other drugs.

Author Response

Reviewer 2

Lim et al. explored the role of NHA and HAD in inhibiting osteoclastogenesis in this manuscript. The topic is quite interesting and the structure of the study is clear and easy to understand. However, I have several suggestions to strengthen the rationality of this manuscript.

Point 1) In the previous article, the author analyzed 10 different compounds why choose the compound 6? For example, the most significant drug in inhibiting ERK pathway activation was compound 5. In addition, what is the structure of HAD, a derivative of NHA, and is there any previous literature discussing its activity?

Response 1) Thanks for your review. We started this study using isolated compounds in previous study. In previous study, we obtained 3.8 mg of compound 5 and 26.2 mg of compound 6. Thus we chose compound 6 (NHA) in this study. And we added structures of NHA and HAD in Figure 1A.

Point 2) Fig.1a, this picture is fuzzy, and it is suggested that the author should place a higher magnification and higher resolution figure for presentation.

Response 2) Thanks for your review. We revised figure 1a.

Point 3) The definition of TRAP-positive multinucleated cell (line 76-77) should be placed in the figure legend.

Response 3) Thanks for your review. We replaced legend to “TRAP-positive multinucleated cells (≥3 nuclei) were counted.”

Point 4) Besides the differentiation of osteoclast as shown in Figure 1, do NHA and HAD have any effect on actin ring formation?

Response 4) Thanks for your comment. Figure 1B data shows only effect of compounds on osteoclast formation. The actin ring usually formed in mature osteoclast and compounds inhibited cell fusions, which are occurred in initial stage of osteoclastogenesis. Thus, to confirm the effect of compounds on actin ring formation, additional experiment is necessary. 

Point 5) In Figure 3A, why is the p-Akt basal level of BMMs so high without RANKL?

Response 5) Thanks for your review. In Figure 3, we didn’t use FBS starvation process. Thus, FBS could cause the high basal level of p-Akt.

Point 6) In FIG. 3B, at 0 h, why are the expressions of all proteins high in the RANKL+HAD group? Is there a group without RANKL as a control group for protein expression?

Response 6) Thanks for your review. At 0h, there is no RANKL+HAD group. 0h is a group that has not processed anything. At 0h, the graph was corrected to black. We are sorry about we didn’t test compounds treatment at 0 h.

Point 7) Alendronate appears in figure 4, but is not explained in the manuscript.

Response 7) Thanks for your comment. We added it in line 122.

Point 8) It is not appropriate to introduce experimental results in detail in the discussion section (Line 189-198...). The authors may describe whether there are other related compounds in clinical use for osteoporosis prevention or the expected effectiveness of the compound discussed here compared to other drugs.

Response 8) Thanks for your review. We revised it as “In the OVX model, NHA and HAD improved bone microarchitecture. It could be explain that the compounds decrease osteoclast formation in bone matrix then, bone resorption of trabecular bone was reduced and bone microarchitecture is recovered. The serum CTX level data shows that the compounds treatment downregulates bone resorption in OVX model. Furthermore, Serum level of RANKL, OPG and RANKL/OPG ratio also indicate that NHA and HAD affect RANKL/OPG axis of osteoblast and inhibit osteoclast formation.”

Reviewer 3 Report

H-J Lee et al indicated a final result that n-trans-hibiscusamide (NHA) and its derivative 4-O-(E)- 18 feruloyl-N-(E)-hibiscusamide (HAD) inhibit ovariectomy-induced osteoporosis by suppressing osteoclast formation. Also, the recovery effect of NHA and HAD was compared with anti-osteoporotic drug alendronate. In the aspect of these results, it should be correctly remarked whether NHA and HAD has the preventive effect or anti-osteoporotic treatment effect. Further, the inhibitory effect of NHA and HAD on in vitro osteoclast differentiation is critical for showing in vivo osteoclast formation on the surface of trabecular bone tissues.

Minor points:

  1. The MS includes Methods in the description of Results and data description is inadequate for bone biology.
  2. The results should be clearly described in all descriptions. For example, in abstracts: At the molecular level, HAD significantly downregulated the phosphorylation of Akt, NF-κB and, mitogen-activated protein kinase (MAPK) signalling molecules. However, NF-κB phosphorylation was not affected by NHA. In results: The phosphorylation of MAPKs, including ERK, JNK and p38, was significantly downregulated by NHA and HAD at one time point at least (Figure 3). However, the phosphorylation of NF-κB p65 and Akt was significantly downregulated only after NHA or HAD treatment. As suggestion, HAD inhibited RANKL-induced NF-kB p65 phosphorylation, but not by NHA. In contrast to NF-kB signaling, NHA blocked RANKL-induced Akt activation, but not by HAD. The combined results indicate that the inhibitory effect of NHA and HAD on osteoclast differentiation may be caused by suppressing distinctive and different targets of RANKL-induced osteoclastogenic signaling.
  3. The MS has error types (e.g. Microunit in Figure 1.) and different font size within each Figure and between Figures.

Author Response

Reviewer 3

H-J Lee et al indicated a final result that n-trans-hibiscusamide (NHA) and its derivative 4-O-(E)- 18 feruloyl-N-(E)-hibiscusamide (HAD) inhibit ovariectomy-induced osteoporosis by suppressing osteoclast formation. Also, the recovery effect of NHA and HAD was compared with anti-osteoporotic drug alendronate. In the aspect of these results, it should be correctly remarked whether NHA and HAD has the preventive effect or anti-osteoporotic treatment effect. Further, the inhibitory effect of NHA and HAD on in vitro osteoclast differentiation is critical for showing in vivo osteoclast formation on the surface of trabecular bone tissues.

Minor points:

Point 1) The MS includes Methods in the description of Results and data description is inadequate for bone biology.

Response 1) Thanks for your comment. Based on your comments, we have revised the contents of the bone in the discussion part (red color).

Point 2) The results should be clearly described in all descriptions. For example, in abstracts: At the molecular level, HAD significantly downregulated the phosphorylation of Akt, NF-κB and, mitogen-activated protein kinase (MAPK) signalling molecules. However, NF-κB phosphorylation was not affected by NHA. In results: The phosphorylation of MAPKs, including ERK, JNK and p38, was significantly downregulated by NHA and HAD at one time point at least (Figure 3). However, the phosphorylation of NF-κB p65 and Akt was significantly downregulated only after NHA or HAD treatment. As suggestion, HAD inhibited RANKL-induced NF-kB p65 phosphorylation, but not by NHA. In contrast to NF-kB signaling, NHA blocked RANKL-induced Akt activation, but not by HAD. The combined results indicate that the inhibitory effect of NHA and HAD on osteoclast differentiation may be caused by suppressing distinctive and different targets of RANKL-induced osteoclastogenic signaling.

Response 2) Thanks for your review. We revised abstract section to “At the molecular level, NHA and HAD significantly downregulated the phosphorylation of mitogen-activated protein kinase (MAPK) signalling molecules. However, Akt and NF-κB phosphorylation was inhibited only after NHA or HAD treatment.”

Point 3) The MS has error types (e.g. Microunit in Figure 1.) and different font size within each Figure and between Figures.

Response 3) Thanks for your review. We revised font style and size of all figures.

Round 2

Reviewer 2 Report

The authors have mostly addressed all my comments.

Author Response

Reviewer 2

The authors have mostly addressed all my comments.

Response) Thanks for your review.

Reviewer 3 Report

The MS has more questions as follows:

  1. The MS  needs the discussion in the aspect of preventive or anti-osteoporotic effect of NHA or HAD.
  2. It should be measured osteoclast number on the surface of trabecular bone tissues.
  3. What is the reasonable evidence to exclude bone formation by osteoblasts in NHA or HAD-mediated anti-osteoporotic event.

Author Response

Reviewer 3

The MS has more questions as follows:

Point 1) The MS needs the discussion in the aspect of preventive or anti-osteoporotic effect of NHA or HAD

Response 1) Thanks for your comment. We added additional text in discussion about preventive or anti-osteoporotic effect of NHA or HAD (Line 203~208).

Point 2) It should be measured osteoclast number on the surface of trabecular bone tissues.

Response 2) Thanks for your review. We also agree with your comment that measuring osteoclast number on the surface of trabecular bone is the best way to show the effect of compounds. However, we showed recovery of bone microarchitectures (Micro-CT data), reduced bone resorption (CTX data) and inhibition of osteoclastogenesis (in vitro). These results could indicate that the compounds inhibit osteoclast differentiation and the number of osteoclasts on the surface of trabecular bone tissues.

Point 3) What is the reasonable evidence to exclude bone formation by osteoblasts in NHA or HAD-mediated anti-osteoporotic event

Response 3) Thanks for your review. This study started from anti IL-6-signaling effect of compounds. It is closely related with anti-osteoclastogenesis. Thus, we could ensure that the compounds could inhibit osteoclast differentiation and focused on it. However, if we didn’t know the anti-IL-6 effect of compounds and we focused on the bone biological effects of the compounds, we would include about bone formation.